

**PeerJ Hubs**
Published on behalf of



# Optimizing *in vitro* fertilization in four Caribbean coral species

Valérie F. Chamberland[1,2,3,*], Matthew-James Bennett[1,*],
Tania Doblado Speck[1], Kelly R. W. Latijnhouwers[1,2,3] and
Margaret W. Miller[1]

[1] SECORE International, Miami, Florida, United States
[2] CARMABI Foundation, Willemstad, Curaçao
[3] Department of Freshwater and Marine Ecology, University of Amsterdam, Amsterdam, Netherlands
* These authors contributed equally to this work.

Corresponding author
Valérie F. Chamberland,
v.chamberland@secore.org

## ABSTRACT

**Background:** Larval propagation and seeding of scleractinian corals for restoration is a rapidly expanding field, with demonstrated applications to assist the recovery of declining populations on reefs. The process typically involves collecting coral reproductive material, facilitating *in vitro* fertilization (IVF), and settling and outplanting the resulting coral offspring. Optimizing IVF can reduce gamete wastage and increase larval yields for propagation, therefore improving the efficiency of this intervention.

**Methods:** In this study we tested three IVF conditions in four Caribbean broadcast-spawning coral species (*i.e.*, *Diploria labyrinthiformis*, *Colpophyllia natans*, *Pseudodiploria strigosa*, *Orbicella faveolata*) to determine sperm concentration, gamete age, and co-incubation time resulting in the highest fertilization success. For each species, we exposed eggs from a single dam to pooled sperm samples from three sires (1) at concentrations ranging from zero to $10^9$ cell $mL^{-1}$, (2) after letting gametes age for 2 to 6 h, and (3) for a period of 15 to 120 min.

**Results:** These experiments revealed a gamete longevity of at least 4 h and clear minimum sperm concentration thresholds ($>10^5$ to $10^6$ cell $mL^{-1}$) in all four species. Fertilization took place much faster than expected ($\leq$15 min) in the three brain corals under study, whereas *O. faveolata* gametes required a co-incubation period of 60 to 120 min to achieve maximum IVF success.

**Discussion:** We present these results in the context of IVF data available for other hermaphroditic broadcast-spawning scleractinians. We then provide recommendations for coral breeding practitioners to maximize larval production from gamete collections, and finally, we discuss our findings' potential implications on fertilization dynamics during natural coral spawning events.

# INTRODUCTION

The rapid and persistent decline of coral reefs worldwide stresses the need for active reef restoration, in concert with effective policies that address local anthropogenic disturbances

and broader climate change effects (*Knowlton et al., 2021*). Interventions that have potential to restore coral abundance and (genetic) diversity on reefs, including sexual larval propagation and assisted gene flow, are progressing but remain limited in scale (*Baums et al., 2019*; *Randall et al., 2020*; *Hagedorn et al., 2021*; *Banaszak et al., 2023*). Improving efficiency and effectiveness in such interventions is vital for the expansion of their application (*Vardi et al., 2021*; *Banaszak et al., 2023*).

For broadcast-spawning coral species, the practice of sexual coral propagation typically involves four sequential phases; first the collection of gametes either from spawn slicks or from individual wild or captive colonies, second, the facilitation of *in vitro* fertilization (IVF) to generate coral embryos (unless fertilization has naturally occurred in a spawn slick), third the culturing of these embryos until they reach larval competency, and fourth the settling and seeding (outplanting) of the resulting coral offspring into the wild (*Randall et al., 2020*; *Banaszak et al., 2023*). Obtaining high fertilization success during IVF ensures that the majority of harvested eggs will yield coral larvae and not go to waste, and minimizes the risk of culture failure associated with water quality issues resulting from the deterioration of unfertilized eggs. Yearly larval production is constrained by the small number of annual coral reproduction events (*Baird, Guest & Willis, 2009*), by the limited capacity of *ex situ* culturing systems that are often expensive to build and/or labor-intensive to run (*Nakamura et al., 2011*; *Chamberland et al., 2015*), and/or by local conditions preempting the use of larger scale, low-tech *in situ* larval rearing mesocosms (*Doropoulos et al., 2019*; *Suzuki et al., 2020*; *Miller et al., 2022*). Therefore, maximizing fertilization success and maintaining water quality at high larval densities has potential to increase the number and quality of coral offspring per spawning event that can be seeded on reefs for restoration.

Currently, coral IVF is exercised following standard practices evolving from a growing body of research since the 1980s. Gametes are ideally sourced from a diverse pool of parental colonies to maximize genetic diversity and adaptive potential in larval cultures (*Baums et al., 2019*, *2022*), and to minimize risks of low fertilization success between poorly compatible mates (*i.e.*, clonal, closely related, and/or genetically incompatible colonies) (*Willis et al., 1997*; *Baums et al., 2013*; *Miller et al., 2018*). Gamete longevity varies among species, but typically, sperm cease swimming and eggs start deteriorating 4 to 8 h post spawning, after which fertilization rates decrease dramatically (*Heyward & Babcock, 1986*; *Oliver & Babcock, 1992*; *Willis et al., 1997*; *Fogarty, Vollmer & Levitan, 2012*; *Chui et al., 2014*; *dela Cruz & Harrison, 2020*). Therefore, male and female gametes are mixed and co-incubated within a few hours after collection, when gamete viability is assumed to be optimal. Early work on coral IVF further established that fertilization success as a function of sperm concentration follows a classic S-shape curve, with fertilization rates typically declining at sperm concentrations below $10^4$ to $10^5$ cell mL$^{-1}$ (*Oliver & Babcock, 1992*; *Willis et al., 1997*; *Levitan et al., 2004*). In more recent years, this pattern was confirmed in many other Caribbean and Indo-Pacific coral species (*Fogarty, Vollmer & Levitan, 2012*; *Nozawa, Isomura & Fukami, 2015*; *dela Cruz & Harrison, 2020*; *Buccheri et al., 2023*). However, inhibition of fertilization has been observed in several species when IVF was conducted at concentrations surpassing $10^7$ cell mL$^{-1}$ (*Oliver & Babcock, 1992*;

*Fogarty, Vollmer & Levitan, 2012*). This may be caused by decreased oxygen, increased $CO_2$, and lowered pH in dense sperm cultures (*Chia & Bickell, 1983*; *Oliver & Babcock, 1992*), and/or to polyspermy, a process by which more than one spermatozoa fuse with a single egg causing the death of the egg (*Yund, 2000*; *Franke, Babcock & Styan, 2002*). Hence, nearly all practitioners seeking to maximize fertilization success target a saturation sperm concentration of ~$10^6$ to $10^7$ cell mL$^{-1}$ based on visual assessments of the mixture's opacity (*Hagedorn et al., 2009*; *Nakamura et al., 2011*; *Chamberland et al., 2015*; *Severati et al., 2024*), and this concentration is now recommended in coral breeding guidelines to yield maximum fertilization while moderating risks of polyspermy and reduced water quality (*Guest et al., 2010*; *Marhaver, Chamberland & Fogarty, 2017*; *Banaszak et al., 2018*). Male and female gametes are typically co-incubated for ~30 to 120 min (*Hagedorn et al., 2009*; *Guest et al., 2010*; *Chamberland et al., 2015*, *2017*; *Marhaver, Chamberland & Fogarty, 2017*) or until eggs initiate a first cleavage, after which zygotes/embryos are gently washed with seawater to discard excess sperm in the culture before they are transferred to either *ex situ* (*e.g.*, *Hagedorn et al., 2009*; *Guest et al., 2014*; *Chamberland et al., 2015*; *Calle-Triviño et al., 2018*; *Sellares-Blasco et al., 2021*; *Mendoza Quiroz et al., 2023*) or *in situ* rearing systems (*e.g.*, *Doropoulos et al., 2019*; *Suzuki et al., 2020*; *Sellares-Blasco et al., 2021*; *Miller et al., 2022*).

While these standard practices oftentimes produce good fertilization rates in broadcast-spawning coral species, they are not universally favorable. Optimum conditions for IVF can vary quite considerably among species and are not always conserved at the genus level. In the Caribbean, *Acropora palmata* eggs are fertilized less readily than sister taxa *A. cervicornis*, requiring an order of magnitude more sperm to achieve maximum fertilization success (*Fogarty, Vollmer & Levitan, 2012*). In the Indo-Pacific species *Favites colemani* and *Platygyra sinensis*, initiating gamete co-incubation 4 h past spawning results in poor fertilization success (<25%), indicating decreased viability (*Oliver & Babcock, 1992*; *dela Cruz & Harrison, 2020*), whereas sperm and eggs produced by *Platygyra acuta* can still yield >60% fertilization success 6.5 h after spawning (*Chui et al., 2014*). In *Favites pentagona* and *F. valensiennesi*, a gamete co-incubation period of 30 to 60 min is required for fertilization to take place, whereas in *A. gemmifera* and *F. abdita* shorter contact times of 15 min are sufficient (*Nozawa, Isomura & Fukami, 2015*). Furthermore, sperm concentration, gamete age, and co-incubation period can act synergistically in influencing fertilization success. For example, exposing eggs to sperm for 10 to 30 s can yield acceptable fertilization success under elevated sperm concentrations, though interactions between sperm concentration and gamete contact time on fertilization success differ across taxa (*Buccheri et al., 2023*). Overall, species-specific studies on the fertilization ecology of scleractinian corals are currently lacking despite their clear applications for IVF, as well as their value in better comprehending fertilization dynamics in the field.

Here, we aimed to improve the efficacy of IVF in the three Caribbean brain coral species *Diploria labyrinthiformis* (Gregory, 1900; Fig. 1A), *Colpophyllia natans* (Houttuyn, 1772; Fig. 1B), and *Pseudodiploria strigosa* (Dana, 1846; Fig. 1C), as well as in one species of the *Orbicella* (formerly *Montastraea*) *annularis* species complex, *O. faveolata* (Ellis and Solander, 1786; Fig. 1D). All are reef-building species that are hermaphroditic and

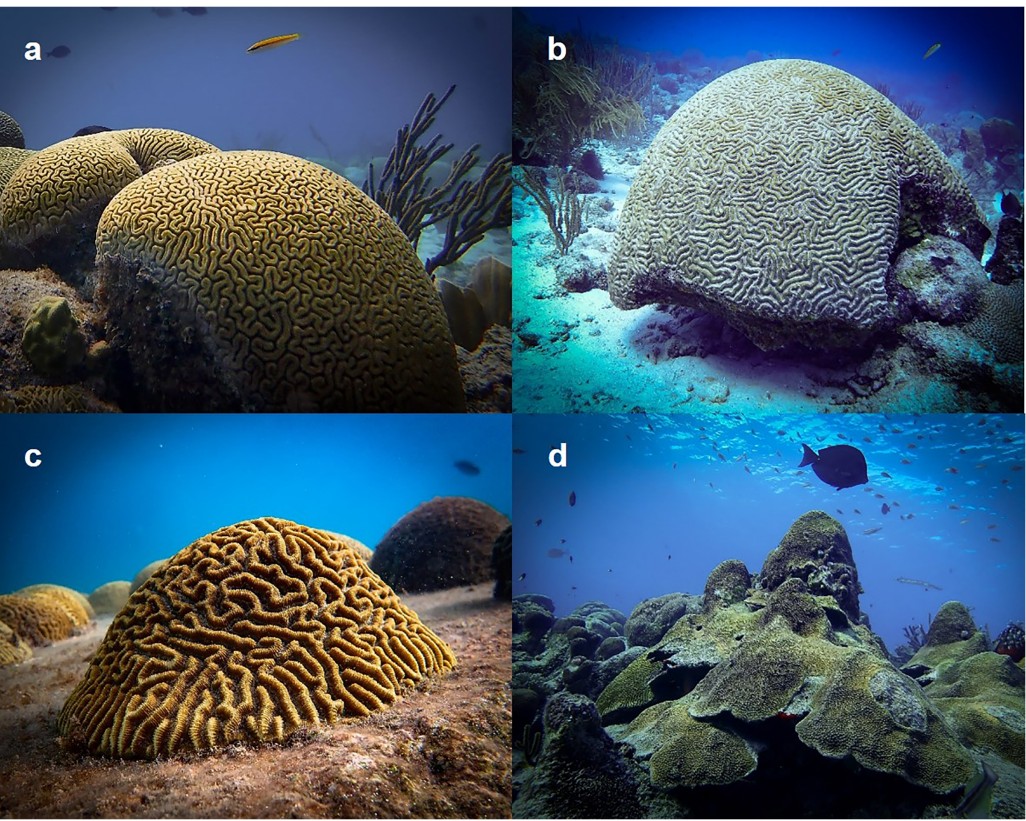

**Figure 1 Study species.** (A) *Diploria labyrinthiformis*, (B) *Colpophyllia natans*, (C) *Pseudodiploria strigosa*, and (D) *Orbicella faveolata*. Gametes collected from these species were used in a series of fertilization assays to quantify the influence of sperm concentration, gamete age, and gamete co-incubation time on IVF success. Photo credits: Lars ter Horst.

reproduce *via* broadcast spawning *i.e.*, they release packets containing both eggs and sperm (egg-sperm bundles) in the water column where fertilization occurs. Detailed information on IVF is lacking for all four species even though they are routinely propagated for population enhancement and restoration purposes (*Chamberland et al., 2017*; *Sellares-Blasco et al., 2021*; *Miller et al., 2022*). We therefore ran three different fertilization assays to identify (1) minimum sperm concentration, (2) optimal gamete age, and (3) best gamete co-incubation time to optimize fertilization success in these species. We then contextualize our findings building on a review on coral IVF by *Nozawa, Isomura & Fukami (2015)* as well as more recent publications, we provide applications for coral breeding practitioners to improve IVF success in these four species, and we discuss our findings' significance for fertilization dynamics during natural coral spawning events.

## MATERIALS AND METHODS

### Gamete collections

We conducted this study on the island of Curaçao in the Southern Caribbean where we collected gametes on predicted spawning nights (*Marhaver, Chamberland & Vermeij, 2024*) in July 2019 (*D. labyrinthiformis*), September 2019 (*P. strigosa*), October 2019

(*C. natans*), and October 2020 (*O. faveolata*). All spawn collections were performed under research and collection permit #2019/021824 granted to the CARMABI Research Station by the Ministry of Health, Environment and Nature of Curaçao. Spawn collection dates and times for each species are available in Table S1. Spawning colonies were tented with cone-shaped nets only once gamete bundles were clearly observed in the polyp mouths and release was imminent. Positively buoyant egg-sperm bundles accumulated in inverted removable 50-mL polypropylene conical centrifuge tube (Falcon; Corning Life Sciences, Tewksbury, MA, USA) secured at the top of these nets. In most cases gamete bundles took more than an hour to break up and release their contents (Table S1). During this time, we were generally able to (1) reduce the volume of seawater inside the collection tubes to a 1:1 bundle:seawater ratio to maintain high sperm concentrations and (2) return to the laboratory. All fertilization assays began within an hour after gamete bundles had broken up. Temperature in the laboratory was maintained between 28 and 29 °C, similar to daily average sea surface temperatures (SSTs) in July, September, and October 2019, and in October 2020 (*NOAA Coral Reef Watch, 2020*).

## Experimental design

A schematic representation of the experimental design is shown in Fig. 2. For all three assays, we selected three parents as sperm-donors and a separate parent as egg-donor (Fig. 2). Hence, in each fertilization assay, eggs from a single dam had the opportunity to be fertilized by sperm from three other, non-self sires (Fig. 2). Individual parent collections with similarly high gamete densities and no visible signs of contamination or oocyte malformation were prioritized for the fertilization assays. We opted for one, single-dam-multi-sire cross (1♀ × 3♂) per species, rather than conducting multiple pairwise, single-dam-single-sire crosses (1♀ × 1♂). This design aimed to minimize risks of individual incompatibilities in pairwise crosses which have been observed in several scleractinian species (*Willis et al., 1997*; *Baums et al., 2013*; *Miller et al., 2018*). We ideally would have incorporated additional biological replicates (*i.e.*, several crosses with unique dams) to capture inter-genet variability among dams of each species. This was however not feasible due to logistical limitations on the nights of spawn collections, namely completing the numerous manipulations required to conduct each fertilization assay within the ~5 h window before gamete quality diminishes consistently across all four species. For each experimental treatment, we did however include technical replicates (*i.e.*, the same cross repeated independently for each experimental treatment) (*n* = 3 for *C. natans*, *P. strigosa*, and *O. faveolata* and *n* = 5 or 6 for *D. labyrinthiformis*).

## Gamete preparation

We separated sperm and eggs upon arrival at the laboratory (Fig. 2). We first transferred each gamete collection into a modified 50-mL conical tube (Falcon; Corning Life Sciences, Tewksbury, MA, USA) fitted with a control valve at its base. Once all egg-sperm bundles had broken up, we drained sperm *via* the control valve, through a 100-μm mesh cell-strainer to remove impurities, and into a clean 50-mL conical tube. Positively buoyant eggs
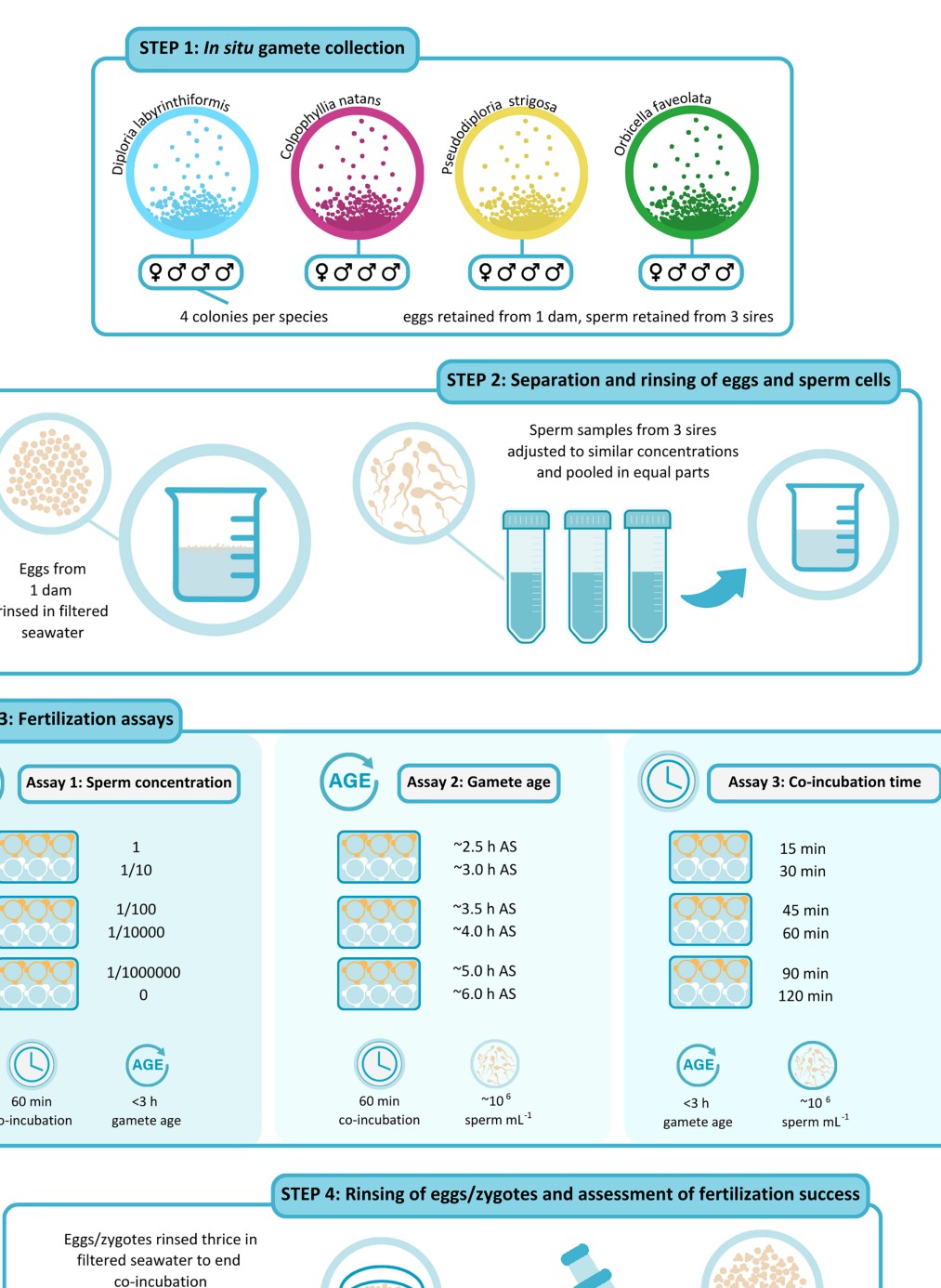

**Figure 2  Schematic representation of the experimental design.** (1) Gametes were collected *in situ* from four colonies per species on predicted nights of spawning. (2) After gamete bundles had broken up, sperm and eggs were separated and washed. Eggs from 1 dam were rinsed in 0.5-μm-filtered seawater (FSW) and kept aside in a 150-mL beaker filled with FSW. Sperm samples from three sires were sieved through 100-μm cell strainers

**Figure 2 (continued)**
and adjusted to similar concentrations using FSW before they were pooled in equal parts in a150-mL plastic beaker. (3) Fertilization assays were carried out in six-well plates ($n$ = 3 to 6 wells per treatment). In Assay 1, ~200 eggs were placed in wells pre-filled with 10 mL of the sperm solution in serial dilutions of 1, 1/10, 1/100, 1/10,000, 1/1,000,000, and 0 at the start of the experiment. Gamete co-incubation was stopped after 60 min. In Assay 2, ~200 eggs were placed in 10 mL of the sperm solution at a concentration of $10^6$ cell mL$^{-1}$ after they had been kept aside for ~2.5, 3, 3.5, 4, 5, and 6 h after spawning. Gamete co-incubation was stopped after 60 min. In Assay 3, ~200 eggs were placed in wells pre-filled with 10 mL of the sperm solution at a concentration of $10^6$ cell mL$^{-1}$ at the start of the experiment. Gamete co-incubation was stopped after 15, 30, 45, 60, 90, and 120 min. (4) At the end of gamete co-incubation, eggs/zygotes were rinsed thrice in FSW, inside cell strainers and were kept separately until scored. When all fertilized eggs had at least reached a 64-cell stage, fertilization success was photo-documented under a dissecting stereomicroscope and was calculated as the proportion of the total number of eggs showing signs of cell cleavage. Illustration by Shira de Koning @ By Shira.

remained in the modified tubes where we rinsed them by repeatedly adding and draining 0.5-μm-filtered seawater (FSW) until clean, after which they were transferred into larger 150-mL plastic beakers filled with 100 mL of FSW. Eggs were stirred occasionally with a clean pipette until they were used in fertilization assays.

We diluted the two densest sperm collections to a concentration approximately equal to that of the most dilute collection by adding FSW until reaching a similar opacity. We then pooled 30 mL of sperm from each of these three sperm donors to ensure each sire was approximately equally represented in the 90-mL sperm pool (Fig. 2). Gamete samples were then assigned to one of three assays aimed at determining fertilization success as a function of (1) sperm concentration, (2) gamete age, and (3) gamete co-incubation time.

In the sperm concentration experiment (Assay 1), 30 mL of this dense sperm mixture were used to perform a series of dilutions as described below (Fig. 2). For use in Assay 2 and Assay 3, we further diluted a 30 mL aliquot of this mixture to target (nominal) $10^6$ cell mL$^{-1}$ using FSW, a concentration previously resulting in high fertilization success in several other coral species (*Nozawa, Isomura & Fukami, 2015*) (Fig. 2). Due to time limitations, it was not possible to quantify sperm concentration before starting the assays. This dilution was therefore achieved based on a visual assessment of the mixture's opacity. Samples of the concentrated and the diluted sperm mixtures were fixed in an 8% solution of glutaraldehyde in FSW to allow *post hoc* quantification of concentration using a haemocytometer chamber (Bright-Line; Hausser Scientific, Horsham, PA, USA) under a phase contrast optical microscope (Leica, Heerbrugg, Switzerland). These actual concentrations are available in Table S2.

## Fertilization assays

**Assay 1: sperm concentration**—This assay aimed to determine the optimal sperm concentration for successful IVF for each of the four dams representing our four study species. We did so by comparing the success of fertilization carried out at five different sperm concentrations (Fig. 2). From the concentrated mixture described above, we produced a series of eight 1/10 dilutions in FSW and selected four of these based on their opacity to target a range of ~$10^2$ to $10^9$ cell mL$^{-1}$ *sensu Nozawa, Isomura & Fukami (2015)* (Table S2a). For example, in *C. natans* the initial pooled sperm sample contained $1.06 \times 10^9$ cell mL$^{-1}$, resulting in five tested conditions of $1.06 \times 10^9$, $10^8$, $10^6$, $10^4$, and

$10^2$ sperm mL$^{-1}$ (Table S2a). To quantify the possible occurrence of parthenogenesis, self-fertilization, and/or sample contamination with non-self sperm before the experiments started, we further included an additional set of replicates as controls for which we added washed eggs to FSW with no added sperm (Fig. 2). Thus, these 'no-sperm' controls ensured that the fertilization rates obtained from our assays reflected true fertilization success resulting from outcrossing under the different conditions tested, though they had obviously been exposed to self-sperm in the short time prior to separation/washing. Gametes were mixed 30 to 45 min after bundles had broken up and co-incubation time was standardized to 60 min across all treatments.

**Assay 2: gamete age**—To determine the maximum gamete age at which high fertilization success can be achieved for each species, we mixed sperm and eggs after they had been kept separate for a controlled amount of time (Fig. 2). The first time point was initiated once egg-sperm bundles had broken up and we had separated and washed the eggs and sperm. This generally coincided with ~2 h after spawning (AS) (Table S1). The following treatments were started ~30, 60, 90, 120, 180, and 300 min later, corresponding to ~2.5, 3, 3.5, 4, 5, and 6 h gamete age (Fig. 2). Gamete co-incubation period was standardized to 60 min and sperm concentration was the same for all age treatments (~$10^6$ cell mL$^{-1}$, Table S2b). Note that the 4 h gamete age treatment was omitted in *C. natans* due to a manipulation error.

**Assay 3: gamete co-incubation period**—To identify the optimal gamete co-incubation period for successful fertilization, we controlled the amount of time eggs and sperm were exposed to one another (Fig. 2). To do so, we placed all eggs in contact with sperm at the same time, 30 to 45 min after egg-sperm bundles had broken up, but rinsed them after 15, 30, 45, 60, 90, and 120 min. Sperm concentration was the same for all co-incubation time treatments (~$10^6$ cell mL$^{-1}$, Table S2c).

In all three assays, 10 mL of sperm solution were added to individual wells in six-well cell culture plates (CellMAX; Biopioneer Inc., San Diego, CA, USA) fitted with 100-μm mesh cell-strainers (SWiSH Premium; Stellar Scientific, Baltimore, MD, USA) (Fig. 2). Fertilization was initiated when placing ~200 eggs into each of the cell-strainers immersed in the sperm solution. We included five or six technical replicates per treatment for *D. labyrinthiformis* and three for all other species. At the end of each assay, we stopped fertilization by rinsing eggs in a series of water baths consisting of 150-mL plastic beakers filled with 100 mL of FSW (Fig. 2). We carefully lifted each cell strainer containing eggs from the culture plate, partially lowered it into a water bath, and gently swirled it to remove residual sperm. We repeated this process three times in new FSW before transferring eggs into a final beaker filled with clean FSW. When all fertilized eggs had at least reached a 64-cell stage (roughly 3 to 4 h after fertilization), we photographed a subset of eggs (mean of 72, range of 10 to 228) from each replicate under a dissecting stereomicroscope (MBS-10; LOMO, St. Petersburg, Russia) and assessed fertilization rates using the resulting images (Fig. 2), whereby we considered eggs showing no signs of cleavage at this point to be unfertilized. Only intact, full-sized embryos or eggs (and not fragments) were considered during photo-analyses.

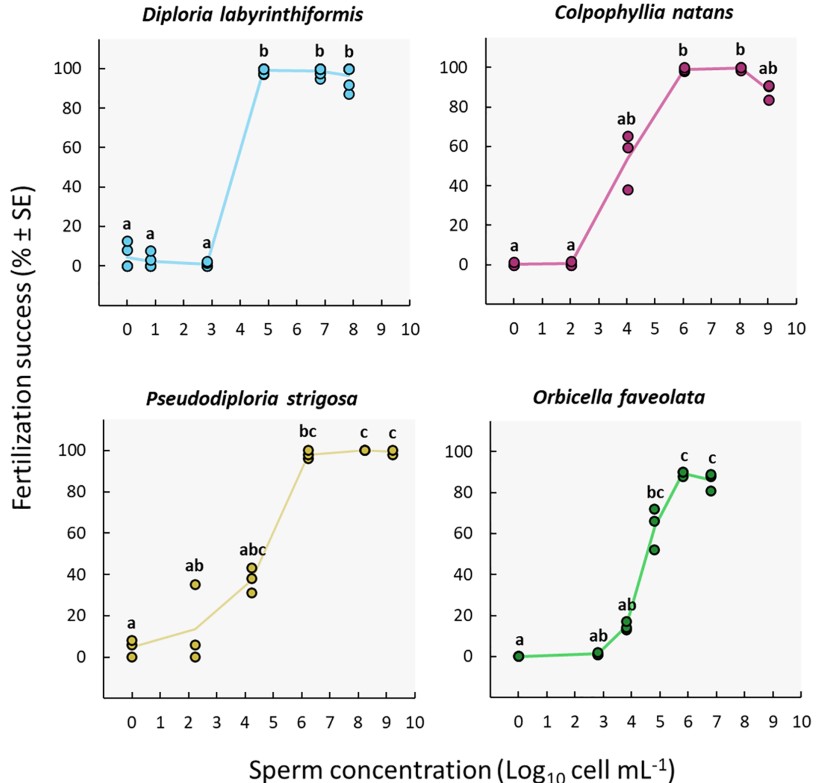

**Figure 3 Fertilization success as a function of sperm concentration.** Eggs from a single dam per species were exposed to a solution of sperm from three sires at concentrations ranging from 0 to ~$10^9$ cell mL$^{-1}$. Gametes were mixed within 3 h after spawning and co-incubated for 60 min. *Points* show data for each individual replicate ($n$ = 3 to 6 per treatment). The *line* connecting average values between each treatment was added to the plots as a visual aid, not for model fitting. *Letters* indicate significantly different groups.

## Data analysis

We performed all statistical analyses in R version 4.2.2 (*R Development Core Team, 2020*). We used one-way Kruskal–Wallis non-parametric tests to determine statistical differences in fertilization success among treatments within each assay. When significant differences were detected, we ran Dunn's *post hoc* pairwise comparisons to identify different treatment groups. All statistical results are available in Table S3.

## RESULTS

### Sperm concentration

Fertilization reached a maximum at sperm concentrations between ~$10^5$ and $10^8$ cell mL$^{-1}$ in *D. labyrinthiformis*, between ~$10^6$ and $10^9$ cell mL$^{-1}$ in *C. natans* and *P. strigosa*, and between ~$10^6$ and $10^7$ cell mL$^{-1}$ in *O. faveolata* (Fig. 3, Tables S2a, and S3a), though for *O. faveolata* we did not test for concentrations above $10^7$ cell mL$^{-1}$. At least 97% of eggs from the three brain coral colonies were fertilized at these concentrations, and fertilization success for *O. faveolata* reached 89% ± 0.7 ($n$ = 3) (mean ± SE) at a

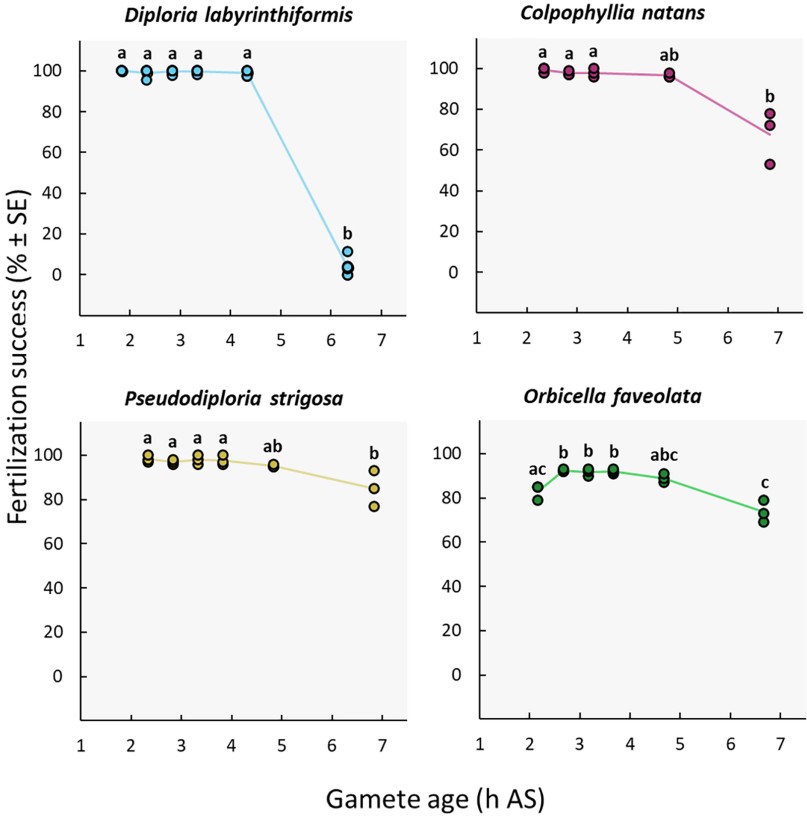

**Figure 4 Fertilization success as a function of gamete age measured in hours after spawning (AS).** Eggs from a single dam per species were exposed to a solution of sperm from three sires after they were left to age for ~2 to 6 h. Gametes were co-incubated for 60 min at a sperm concentration of $10^6$ sperm mL$^{-1}$. *Points* show data for each individual replicate ($n$ = 3 to 6 per treatment). The *line* connecting average values between each treatment was added to the plots as a visual aid, not for model fitting. *Letters* indicate significantly different groups.

concentration of $6.4 \times 10^5$ cell mL$^{-1}$. At $10^4$ cell mL$^{-1}$ or less, fertilization almost entirely failed in *D. labyrinthiformis* (<5%), and was below 55% in dams of all three other species (Fig. 3, Tables S2a, and S3a).

We obtained the highest sperm concentrations in collections from *P. strigosa* and *C. natans* (~$10^9$ cell mL$^{-1}$) (Table S2a). While maximum fertilization was obtained at this concentration for *P. strigosa* (99% ± 1 ($n$ = 3)), fertilization success was slightly (albeit not significantly) lower at this concentration in *C. natans* (89% ± 2 ($n$ = 3)) compared to the next highest concentration of $10^8$ cell mL$^{-1}$ (100% ± 1 ($n$ = 3)) (Fig. 3, Tables S2a, and S3a).

None of the control replicates where eggs were rinsed and kept in FSW without sperm presented significant levels of fertilization (≤5%) (Fig. 3, Tables S2a, and S3a). Therefore, parthenogenesis, self-fertilization, and sample contamination across all other experimental conditions were considered negligible.

## Gamete age

Near complete fertilization (95–100%) was achieved in the three brain coral crosses when gametes were combined between 2h20 and 4h20 AS (Fig. 4, Table 1, Tables S2b, and S3b).

**Table 1 Available information on maximum gamete age for successful fertilization in different scleractinian coral species.** Experimental conditions (*i.e.*, sperm concentration and gamete co-incubation period) are provided where available. AS stands for after spawning.

| Family | Species | Maximum gamete age for high fertilization success (h AS) | Gamete age at which fertilization success declines significantly (h AS) | Source | Experimental conditions |
|---|---|---|---|---|---|
| **Acroporidea** | | | | | |
| | *Acropora millepora* | 4h00 (>85%) | 6h00 (<25%) | *dela Cruz & Harrison (2020)* | $10^5$ sperm mL$^{-1}$, 60 min co-incubation |
| | *Acropora millepora* | 6h00 (>90%) | 8h00 (<25%) | *Willis et al. (1997)* | – |
| | *Acropora tenuis* | 5h00 | 7h00 | *Heyward & Babcock (1986)* | Fertilization success not measured |
| | *Acropora tenuis* | 4h00 (>85%) | 6h00 (<5%) | *dela Cruz & Harrison (2020)* | $10^5$ sperm mL$^{-1}$, 60 min co-incubation |
| | *Montipora digitata* | 6h00 | 7h00 | *Heyward & Babcock (1986)* | Fertilization success not measured |
| | *Montipora digitata* | 2h30 (>65%) | – | *Oliver & Babcock (1992)* | $10^6$ sperm mL$^{-1}$, 180 min co-incubation |
| **Merulinidae** | | | | | |
| | *Favites pentagona* | 2h00 (>75%) | – | *Oliver & Babcock (1992)* | $10^5$ sperm mL$^{-1}$, 180 min co-incubation |
| | *Favites colemani* | 2h00 (>90%) | 4h00 (<25%) | *dela Cruz & Harrison (2020)* | $10^5$ sperm mL$^{-1}$, 60 min co-incubation |
| | *Goniastrea aspera* | 6h00 | 7h00 | *Heyward & Babcock (1986)* | Fertilization success not measured |
| | *Goniastrea favulus* | 6h00 | 7h00 | *Heyward & Babcock (1986)* | Fertilization success not measured |
| | *Orbicella (Montastraea) faveolata* | 4h40 (>80%) | 6h40 (<75%) | This study | $10^6$ sperm mL$^{-1}$, 60 min co-incubation |
| | *Platygyra acuta* | 6h30 (>60%) | 7h30 (<50%) | *Chui et al. (2014)* | $10^6$ sperm mL$^{-1}$, 240 min co-incubation |
| | *Platygyra sinensis* | 2h00 (<90%) | 4h00 (<25%) | *Oliver & Babcock (1992)* | $10^5$ sperm mL$^{-1}$, 180 min co-incubation |
| **Mussidae** | | | | | |
| | *Colpophyllia natans* | 4h50 (>95%) | 6h50 (<70%) | This study | $10^6$ sperm mL$^{-1}$, 60 min co-incubation |
| | *Diploria labyrinthiformis* | 4h20 (>95%) | 6h20 (<5%) | This study | $10^6$ sperm mL$^{-1}$, 60 min co-incubation |
| | *Pseudodiploria strigosa* | 4h50 (>95%) | 6h50 (<90%) | This study | $10^6$ sperm mL$^{-1}$, 60 min co-incubation |

In *D. labyrinthiformis*, 99% ± 1 ($n = 6$) of eggs were fertilized in the 4h20 AS treatment, relative to 4% ± 2 ($n = 6$) when gametes were introduced 6h20 AS, suggesting that gametes produced by *D. labyrinthiformis* ceased to be viable within that time window (Fig. 4, Table 1, Tables S2b, and S3b). Fertilization success declined more gradually between 4h40 and 6h50 AS in both *C. natans* and *P. strigosa*, with 68% ± 8 ($n = 3$) and 85% ± 5 ($n = 3$) of eggs from these dams being fertilized when mixed 6h50 AS, respectively (Fig. 4, Table 1, Tables S2b, and S3b).

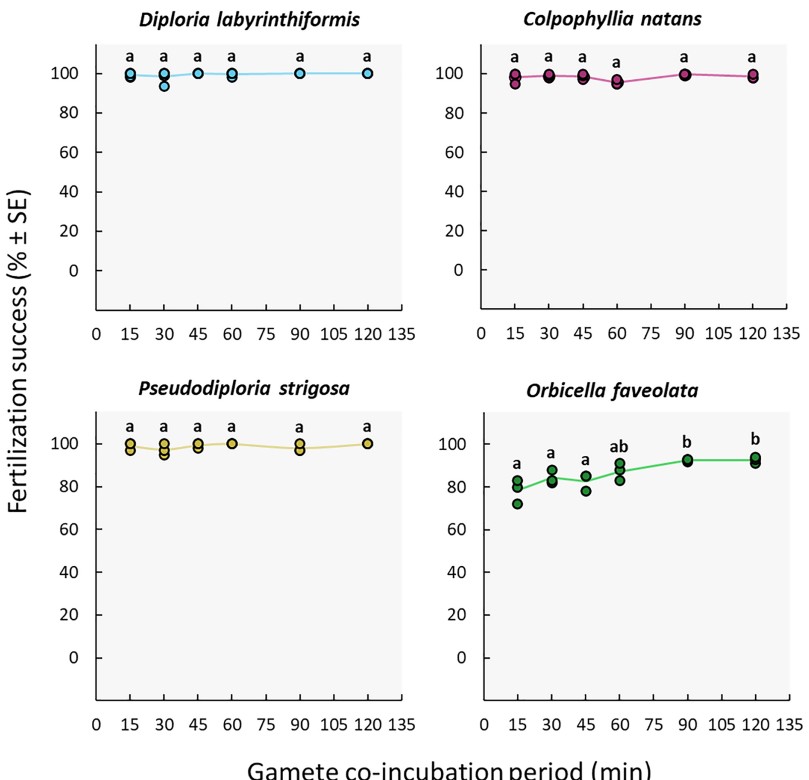

**Figure 5 Fertilization success as a function of gamete co-incubation period, measured in minutes from gamete mixing until eggs were rinsed.** Eggs from a single dam per species were exposed to a solution of sperm for 15, 30, 45, 60, 90 and 120 min before eggs were rinsed. Gametes were mixed within 3 h after spawning and co-incubated at a sperm concentration of $10^6$ sperm $mL^{-1}$. *Points* show data for each individual replicate ($n = 3$ to 6 per treatment). The *line* connecting average values between each treatment was added to the plots as a visual aid, not for model fitting. *Letters* indicate significantly different groups.

We observed a different pattern in *O. faveolata*, whereby a delay AS was required to obtain maximum fertilization success. In this dam, when gametes were mixed 2h10 AS, 83% ± 2 ($n = 3$) of eggs were successfully fertilized compared to 89–92% when gametes were aged 2h40 to 4h40 AS (Fig. 4, Table 1, Tables S2b, and S3b). Similar to *C. natans* and *P. strigosa*, a 15% decline in fertilization was observed between 4h40 and 6h40 AS for *O. faveolata*, with 74% ± 3 ($n = 3$) of eggs fertilized at this last time point (Fig. 4, Table 1, Tables S2b, and S3b).

## Gamete co-incubation period

The duration of gamete co-incubation did not influence fertilization success in the *D. labyrinthiformis*, *C. natans* nor *P. strigosa* colonies under study. Maximum fertilization was reached in all conditions, including co-incubation times as short as 15 min (≥97%) (Fig. 5, Tables S2c, and S3c). In contrast, *O. faveolata* gametes required a co-incubation period of 1h to 2h to achieve similar fertilization success rates (87–93%) (Fig. 5, Tables S2c, and S3c).

## DISCUSSION

This study provides new information on the influence of sperm concentration, gamete age, and gamete co-incubation period on the fertilization success of four Caribbean coral species (three mussids and one merulinid species) that are routinely propagated by restoration practitioners. All three factors tested had important repercussions on IVF success. Our results provide useful applications for more effective larval production during coral breeding, as well as insights into fertilization dynamics in the wild during natural coral spawning events.

### Sperm concentration

Sperm concentration was a strong driver of fertilization success in dams of all four species, with clear minimum thresholds to achieve the highest fertilization potential, namely $\sim 10^5$ cell mL$^{-1}$ in *Diploria labyrinthiformis* and $\sim 10^6$ cell mL$^{-1}$ in *Colpophyllia natans*, *Pseudodiploria strigosa*, and *Orbicella faveolata* (Fig. 3). In all crosses, fertilization success decreased dramatically at $\sim 10^4$ cell mL$^{-1}$ or less (Fig. 3). These findings are consistent with previous reports of IVF success in other hermaphroditic broadcast-spawning scleractinians, requiring high sperm concentrations in the range of $10^5$–$10^7$ cell mL$^{-1}$ (Fig. 6, Table S4). In fact, in 21 of 26 species for which information on sperm density and IVF is available, crosses conducted at concentrations below $10^4$ cell mL$^{-1}$ resulted in poor fertilization success (<50%) (Fig. 6, Table S4). Exceptions include *Acropora latistella*, *A. tenuis*, *Favites colemani*, and *Platygyra daedalea*, all of which can maintain intermediate (50–75%) to high (>75%) fertilization success under much lower sperm densities of $10^3$–$10^4$ cell mL$^{-1}$ (Fig. 6, Table S4). Such large differences in effective sperm concentrations could be driven by various, species-specific characteristics such as differential chemoattraction of sperm towards egg chemical signals (*Morita et al., 2006*), varying levels of gamete incompatibilities (*Willis et al., 1997*; *Baums et al., 2013*; *Miller et al., 2018*), as well as variations in egg size (*Levitan, 2006*). In other marine invertebrate species, larger eggs require fewer sperm encounters to achieve fertilization (*e.g.*, sea urchins: *Levitan, 1996*; *Rahman & Uehara, 2004*), bivalves: (*Luttikhuizen, Honkoop & Drent, 2011*), gastropods: (*Huchette et al., 2004*), providing a larger physical target for sperm in the water column. The four species included in this study produce eggs in the size range of $\sim 300$ to 400 µm (Table S1) and exhibited similar sperm concentration requirements for successful fertilization during IVF. Further empirical studies are needed to determine the relationship between sperm limitation and variation in egg size in corals.

Previous studies have reported declining fertilization success attributed to polyspermy or reduced water quality at higher sperm concentrations, including for *A. cervicornis*, *O. franksi*, *M. digitata*, and *F. pentagona* (Fig. 6, Table S4). Here, we found no clear evidence for depressed fertilization at higher sperm concentrations as fertilization success remained high at concentrations equal or above $10^8$ cell mL$^{-1}$ for our crosses with *P. strigosa* (99%), *D. labyrinthiformis* (97%), and *C. natans* (99%), though in *C. natans* fertilization success decreased slightly (albeit not significantly) between $10^8$ and $10^9$ cell mL$^{-1}$ (Fig. 3). These results suggest effective mechanisms preventing polyspermy in eggs of these brain coral

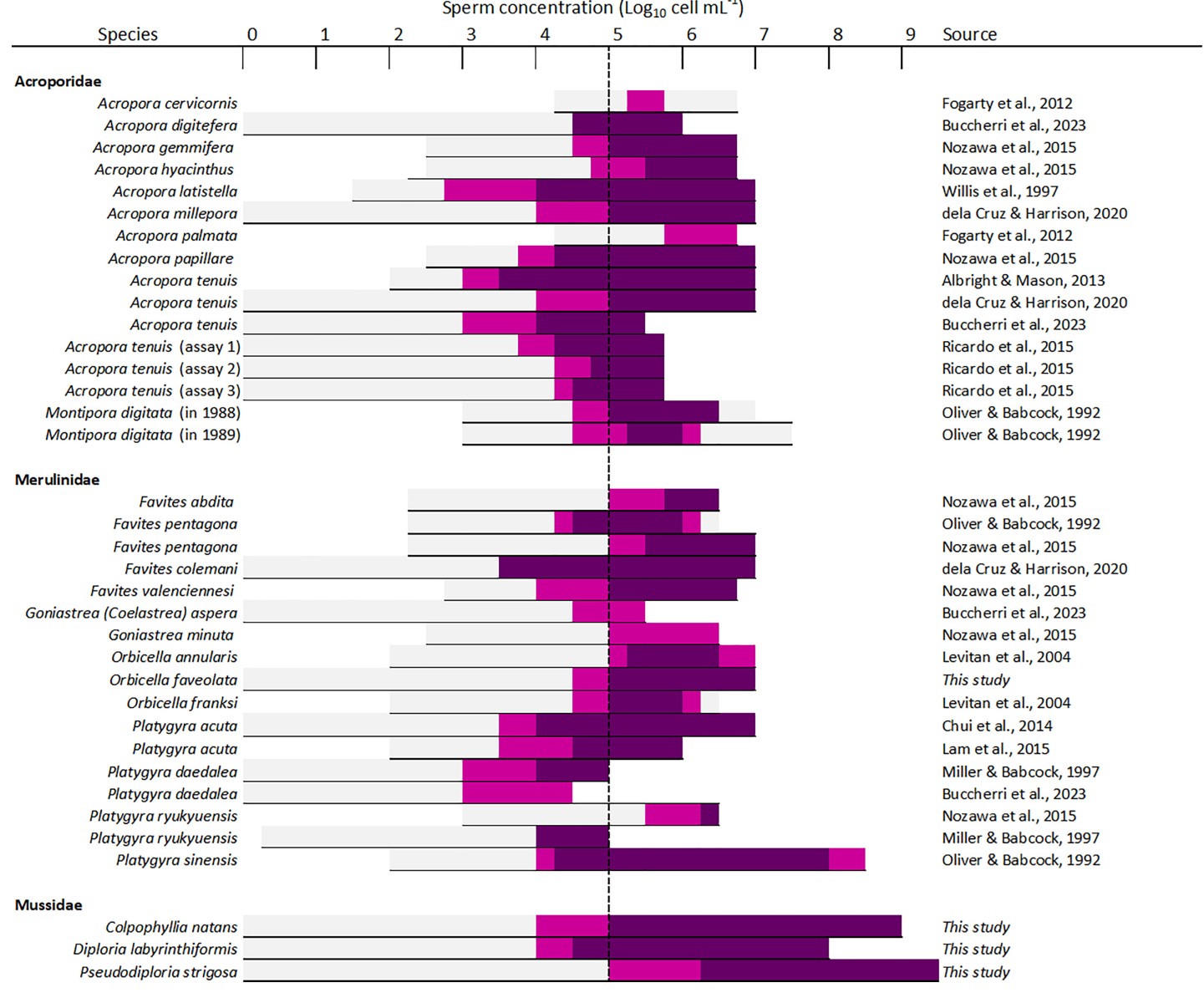

**Figure 6 Overview of available information on the influence of sperm concentration on the fertilization success of scleractinian corals.** The *horizontal line* depicts the range of sperm concentrations that were assessed. The *dark magenta bars* denote sperm concentrations that resulted in >75% fertilization success, *light magenta bars* sperm concentrations that resulted in 50–75% fertilization success, and *light grey bars* sperm concentrations resulting in <50% fertilization success. The *vertical dashed line* indicates the sperm concentration required to achieve ≥50% fertilization success ($10^5$ sperm mL$^{-1}$) in most species (24 out of 26 species). All data supporting this figure is available in Table S4 (*Fogarty, Vollmer & Levitan, 2012*; *Buccheri et al., 2023*; *Nozawa, Isomura & Fukami, 2015*; *Willis et al., 1997*; *dela Cruz & Harrison, 2020*; *Albright & Mason, 2013*; *Ricardo et al., 2015*; *Oliver & Babcock, 1992*; *Levitan et al., 2004*; *Chui et al., 2014*; *Lam et al., 2015*; *Miller & Babcock, 1997*).

species (*i.e.*, fast-block *via* egg membrane depolarization and/or slow-block *via* cortical granule reaction) (*Gould & Stephano, 2003*). Overall, risks of fertilization failure appear more common when cultures are too diluted rather than too dense. Therefore, in accordance with previously established guidelines for coral breeding and restoration

practitioners (*Guest et al., 2010*; *Marhaver, Chamberland & Fogarty, 2017*; *Banaszak et al., 2018*; *Severati et al., 2024*), we recommend conducting IVF with sperm mixtures no more dilute than $10^6$ cell mL$^{-1}$ when possible for these species. Our results further support the likelihood of sperm limitation on sexual coral reproduction *in situ*, whereby gamete dilution may limit the fertilization potential of coral eggs during natural spawning events (*Oliver & Babcock, 1992*; *Levitan et al., 2004*; *Mumby et al., 2024*). Concentrations as high as achieved here ($\geq 10^8$ cell mL$^{-1}$) are unlikely to occur in the open ocean where sperm rapidly disperse in the water column. Thus, risks associated with polyspermy may be marginal in the natural environment relative to *ex situ* fertilization cultures carried out in small volumes of water.

## Gamete age

Our results suggest that combining sperm and eggs as quickly as possible is not essential in *D. labyrinthiformis*, *C. natans*, and *P. strigosa*, and is not recommended in *O. faveolata*. In our *O. faveolata* cross, a delay of 2h40 AS before mixing male and female gametes was required to obtain optimal fertilization (>90%, Fig. 4). *Favites pentagona* displays a similar pattern whereby fertilization success increases gradually, from ~75% to 90%, between 30 min and 2 h after spawning (*Oliver & Babcock, 1992*) (Table 1). In these species, it is possible that fertilization is contingent upon a delayed, final maturation division of the eggs and their release of polar bodies (*Heyward & Babcock, 1986*), or that maximum sperm flagellar motility is achieved after a longer period of time following spawning and/or requires a longer exposure to activation factors from eggs (*Morita et al., 2006*), though we did not record sperm activity in this study. Thus, when *O. faveolata* gametes are mixed soon after spawning, allowing a longer co-incubation period would be recommended, during which time more gametes could become competent.

The extended gamete viability (>4 h) observed in the current study may increase chances of egg and sperm encounters *in situ* under conditions of low sperm availability, or during imperfectly synchronized spawning events by allowing more time for gametes to meet (*Chui et al., 2014*). Together, these findings present restoration opportunities whereby gametes could be collected and later combined from sites representing more distant and distinct habitat types, thus enhancing potentially adaptive genetic diversity in larval cultures (*Baums et al., 2019*, *2022*).

## Co-incubation period

The high success achieved with co-incubation times as short as 15 min indicates that fertilization takes place quickly in *D. labyrinthiformis*, *C. natans*, and *P. strigosa* (Fig. 5) at the egg: sperm ratios tested here. Co-incubation times of 10 min have also yielded adequate fertilization success in other species including *A. digitifera*, *A. papillare*, *A. tenuis*, *Goniastrea aspera*, *Platygyra daedalea*, and *P. ryukyuensis* (*Nozawa, Isomura & Fukami, 2015*; *Buccheri et al., 2023*). Shorter co-incubation periods in larval propagation pipelines allow for earlier washing of the eggs following the mixing of gametes, before they undergo mitotic divisions and hence remain less susceptible to cell breakage during handling

(*Heyward & Negri, 2012*; *Severati et al., 2024*). In contrast, *O. faveolata* gametes required a co-incubation period of 1h to 2h to achieve the highest fertilization rates (87–93%) (Fig. 5), though this could be due to a portion of the gametes needing more time to become competent (see Assay 2: gamete age).

Short *in vitro* co-incubation periods could be a function of egg-regulated sperm motility. Concentrating gametes in this way could increase the exposure of sperm to chemo-attractants that stimulate flagellar activity and aid sperm in locating an egg. Coral eggs also release inhibitors that reduce sperm motility once fertilized (*Coll et al., 1994*; *Morita et al., 2006*; *Morita, Iguchi & Takemura, 2009*). Thus, differing minimum co-incubation periods across species could result from varying levels of sperm motility and chemotaxis, influencing the time required for sperm to locate, swim towards, and fertilize eggs. Shorter periods required for fertilization to take place could have evolved so that fertilized eggs are able to immobilize sperm attempting to penetrate them before the physical interaction causes lethal damage, or to reduce the risks of polyspermy (*Oliver & Babcock, 1992*; *Franke, Babcock & Styan, 2002*). Alternatively, species-specific differences in the 'fertilizability' of eggs such as differences in the egg mucous coat or other processes following bundle breakup might affect required co-incubation time.

Broadcast spawning coral species release large quantities of gametes with a high degree of synchrony and this enhances chances of compatible gamete encounters. The longer the contact time required between sperm and egg for fertilization to take place, the higher the likelihood that gametes will disperse before fertilization has occurred, especially when strong ocean currents occur (*Oliver & Babcock, 1992*; *Levitan et al., 2004*; *Mumby et al., 2024*). Thus, rapid fertilization of eggs in the field may be an adaptation to facilitate fertilization in situations where sperm limitation is likely. Trade-offs may result, however, by reducing the effective number of potential mates that are slightly more distant.

## Possible interactions between factors

In this study we examined three factors separately, holding the other two factors constant at an expected effective level. In practical situations applying IVF, there are likely additional factors that may affect fertilization success. For example, gametic incompatibilities between individual parents have been demonstrated in some species, including *O. faveolata* (*Miller et al., 2018*). For this reason, the effective sperm concentration (*i.e.*, of compatible sperm) in a particular cross may be less than the absolute sperm concentration, and it is possible that this may account for the somewhat lower fertilization levels achieved here for *O. faveolata* overall. Maximizing the number of possible genet pairings in a fertilization cross may help alleviate the risks of fertilization failure due to genetic incompatibilities, though spawn collections from many egg and sperm donors might not always be feasible (*Baums et al., 2022*). Nonetheless, the best mitigation in cases of sperm limitation is likely to extend the co-incubation time, as basic kinetics indicate that more contacts between compatible egg-sperm pairs will occur over time, and no negative effects of extended, two-hour co-incubation times were observed here. For example, observations with *Acropora palmata*, a species with significant

**Table 2 Available information on gamete co-incubation periods achieving over 75% fertilization success in different scleractinian coral species.** Experimental conditions (*i.e.*, sperm concentration and gamete age) are provided where available. AS stands for after spawning.

| Family | Species | Gamete co-incubation period >75% (min) | Source | Experimental conditions |
|---|---|---|---|---|
| **Acroporidea** | | | | |
| | *Acropora digitifera* | 60–90 | Iguchi et al. (2009) | $10^6$ sperm mL$^{-1}$ |
| | *Acropora digitifera* | 10–30 | Buccheri et al. (2023) | $10^4$–$10^5$ sperm mL$^{-1}$, <2 h AS gamete age |
| | *Acropora gemmifera* | 60 | Nozawa, Isomura & Fukami (2015) | $10^6$–$10^7$ sperm mL$^{-1}$, <3 h AS gamete age |
| | *Acropora papillare* | 10–60 | Nozawa, Isomura & Fukami, 2015 | $10^6$–$10^7$ sperm mL$^{-1}$, <3 h AS gamete age |
| | *Acropora tenuis* | 10–30 | Buccheri et al. (2023) | $10^3$ sperm mL$^{-1}$, <2 h AS gamete age |
| **Merulinidae** | | | | |
| | *Favites abdita* | 30–60 | Nozawa, Isomura & Fukami (2015) | $10^6$–$10^7$ sperm mL$^{-1}$, <3 h AS gamete age |
| | *Favites pentagona* | 60 | Nozawa, Isomura & Fukami (2015) | $10^6$–$10^7$ sperm mL$^{-1}$, <3 h AS gamete age |
| | *Favites valenciennesi* | 30–60 | Nozawa, Isomura & Fukami (2015) | $10^6$–$10^7$ sperm mL$^{-1}$, <3 h AS gamete age |
| | *Goniastrea (Coelastrea) aspera* | 10–30 | Buccheri et al. (2023) | $10^6$–$10^7$ sperm mL$^{-1}$, <3 h AS gamete age |
| | *Orbicella (Montastraea) faveolata* | 60–120 | This study | $10^6$ sperm mL$^{-1}$, <3 h AS gamete age |
| | *Platygyra daedalea* | 10–30 | Buccheri et al. (2023) | $10^3$ sperm mL$^{-1}$, <2 h AS gamete age |
| | *Platygyra ryukyuensis* | 10–60 | Nozawa, Isomura & Fukami (2015) | $10^6$–$10^7$ sperm mL$^{-1}$, <3 h AS gamete age |
| **Mussidae** | | | | |
| | *Colpophyllia natans* | 15–120 | This study | $10^6$ sperm mL$^{-1}$, <3 h AS gamete age |
| | *Diploria labyrinthiformis* | 15–120 | This study | $10^6$ sperm mL$^{-1}$, <3 h AS gamete age |
| | *Pseudodiploria strigosa* | 15–120 | This study | $10^6$ sperm mL$^{-1}$, <3 h AS gamete age |

individual incompatibilities (*Baums et al., 2013*; VF Chamberland, 2019, personal observation), suggest that fertilization success at very low sperm concentrations ($\leq 10^3$ cell mL$^{-1}$) can be improved with longer contact times (overnight; ~6 h) (MW Miller, 2014, personal observation). Other complex interactions between sperm concentration and gamete contact time have recently been demonstrated in several Indo-Pacific species. For instance, in *A. digitifera*, 10 min are required for sufficient sperm to find and fertilize eggs under concentrations between $10^3$ and $10^4$ cell mL$^{-1}$ (Table 2), whereas much shorter contact times of 10 to 60 s can yield acceptable fertilization success at $10^5$ cell mL$^{-1}$ in this species (*Buccheri et al., 2023*). Significant within-species variation in fertilization success among different gamete collections or cohorts has also been observed. For example, fertilization success for *Platygyra ryukyuensis* surpassed >75% at Magnetic Island, Australia when conducting crosses at concentrations of $10^4$ sperm mL$^{-1}$ (*Miller & Babcock, 1997*), whereas two orders of magnitude more sperm was required to achieve similar success with spawn collections from Green Island, Taiwan (Fig. 6, Table S4). The current study did not capture potential intra-specific, spatio-temporal variation, since only a single spawn collection was assayed for each species.

Extrapolating our findings to the reef environment requires caution given the multitude of parameters influencing coral sperm and egg encounter rates and fertilization success in the open ocean, which were absent in our assays (*e.g.*, predation: *Westneat & Resing (1988)*, water motion: *Riffell & Zimmer (2007)*, sedimentation: *Ricardo et al. (2015)*, acidification: *Albright & Mason (2013)*). Nonetheless, it is clear that with shrinking coral populations, potential mates have become more and more distant on reefs and that sperm limitation may now prevail during natural spawning events, thus increasing risks of reproductive failure through Allee effects (*Mumby et al., 2024*). All four species under study have suffered significant recent population declines, three of which are currently listed as Critically Endangered (*IUCN, 2022*). Thus, sperm densities are likely to fall below the $10^4$ cell mL$^{-1}$ threshold established in this study by the time eggs meet conspecific sperm, which may require greater travel distances and increased gamete aging, further raising risks of fertilization failure.

## CONCLUSIONS

While our fertilization assays were not designed to capture intra-specific variability in egg characteristics ($n = 1$ dam per species as biological replicate) nor among different gamete collections or cohorts ($n = 1$ spawning event per species), our results nonetheless provide useful, general guidelines for manipulating three factors that impact IVF success *D. labyrinthiformis*, *C. natans*, *P. strigosa*, and *O. faveolata*. We have identified minimum sperm concentration thresholds ranging from $10^5$ to $10^6$ in all four species, and have shown that shorter gamete co-incubation times (15 min) can produce maximum fertilization success in brain corals (but not in *O. faveolata*). This reduces risks of gamete wastage caused by excessive handling and damage to developing embryos when washing them in early stages of cell divisions, after prolonged co-incubation periods. We further found that coral gametes can remain viable over 4 h AS in all four species, opening the possibility for interventions that combine gametes from multiple (distant) sites to boost genetic diversity. While our results, together with previous studies, may support the likelihood of sperm limitation on sexual coral reproduction *in situ*, we show that IVF has the potential to overcome this barrier in dispersed and disconnected coral populations where gametes can be collected and concentrated in a laboratory to achieve high fertilization success. In conclusion, this study highlights variation in fertilization-related traits among Caribbean spawning corals, and the need to tailor protocols to the chosen study species to achieve optimal fertilization success. Expanding the body of knowledge on how to optimize IVF for more coral species will enhance the capacity and efficacy of coral breeding and restoration initiatives.

## ACKNOWLEDGEMENTS

We thank Mark Vermeij and Kristen Marhaver for helping with the 2019 and 2020 coral gamete collection efforts leading to this study. We further thank Ramesh Chatragadda and four anonymous reviewers for providing valuable suggestions to improve earlier versions of this work.

### Funding

Support was provided by The Builders Initiative, The Nature Conservancy, the California Academy of Sciences, and the US National Oceanic and Atmospheric Administration (NA19NMF0080078) through grants to SECORE International. The funders had no role in study design, data collection and analysis, decision to publish, or preparation of the manuscript.

### Grant Disclosures

The following grant information was disclosed by the authors:
SECORE International: NA19NMF0080078.

### Competing Interests

The authors declare that they have no competing interests.

### Author Contributions

- Valérie F. Chamberland conceived and designed the experiments, performed the experiments, analyzed the data, prepared figures and/or tables, authored or reviewed drafts of the article, and approved the final draft.
- Matthew-James Bennett conceived and designed the experiments, performed the experiments, analyzed the data, prepared figures and/or tables, authored or reviewed drafts of the article, and approved the final draft.
- Tania Doblado Speck performed the experiments, prepared figures and/or tables, and approved the final draft.
- Kelly R. W. Latijnhouwers performed the experiments, authored or reviewed drafts of the article, and approved the final draft.
- Margaret W. Miller conceived and designed the experiments, authored or reviewed drafts of the article, and approved the final draft.

### Ethics

The following information was supplied relating to ethical approvals (*i.e.*, approving body and any reference numbers):

The Ministry of Health, Environment and Nature of Curaçao granted research and collecting permits to the CARMABI Research Station enabling this research (permit #2019/021824).

### Data Availability

The raw fertilization rates per replicate for each treatment and for each fertilization assay are available for each of the four study species in the Supplemental Files.

## Supplemental Information

Supplemental information for this article can be found online at http://dx.doi.org/10.7717/peerj.18918#supplemental-information.

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
