# Peer review of "Optimizing in vitro fertilization in four Caribbean coral species"

_PeerJ, doi:10.7717/peerj.18918_

## Round 0.1 · original submission · Minor Revisions

Dear Dr. Chamberland,

Your paper has been reviewed by two experts in the field. They agree that your research was well-executed, and your manuscript provides relevant information to improve Caribbean reef restoration. However, they also provided important suggestions that I hope you address. After you revise the manuscript following the reviewer's suggestions, I will be pleased to reconsider the manuscript for publication in PeerJ. Please make sure to acknowledge the reviewer's valuable contributions to the revised version.

·

Basic reporting

The researchers optimized in vitro fertilization (IVF) conditions for four Caribbean coral species to improve larval propagation for reef restoration. They found:

1.Gametes remain viable for at least 4 hours
2.Minimum sperm concentration thresholds for fertilization
3.Fertilization occurs quickly (within 15 minutes) in some species, while others require longer co-incubation times

These findings provide recommendations for coral breeding practitioners to maximize larval production and aid reef recovery.

Experimental design

This study investigated optimal IVF conditions for four Caribbean coral species by testing:

1. Sperm concentration (0 to 10^9 cells/mL)
2. Gamete age (2 to 6 hours)
3. Co-incubation time (15 to 120 minutes)

The goal was to determine the combination of conditions resulting in the highest fertilization success for each species. The methodology reported here is satisfactory.

Validity of the findings

I validated the findings of this study was assessed based on the following aspects:

1. Consistency: The results show consistent patterns across three out of four species (brain corals), indicating a reliable finding.
2. Specificity: The study identified specific sperm concentration thresholds (>10^5-10^6 cells/mL) and co-incubation times (15 min to 120 min) for each species, adding to the validity.
3. Comparison to existing knowledge: The discussion contextualizes the results within the existing body of coral IVF research, suggesting that the findings align with or build upon previous studies.
4. Recommendations: The provision of recommendations for coral breeding practitioners implies that the results have practical applications, increasing their validity.
5. Limitations: The study only investigated four species, and the results might not be generalizable to all coral species. However, this is acknowledged in the discussion, which strengthens the validity.

Overall, the findings appear to be valid, with consistent and specific results that align with existing knowledge and have practical applications. However, further research could enhance generalizability and confirm the results in other coral species.

Additional comments

The article presents sufficient data to be considered acceptable in this field of research. However, I recommend revising certain sentences and wording to employ more precise scientific terminology in place of general language. Additionally, a thorough review of the text is necessary to correct grammatical errors and enhance overall clarity.

Best wishes

Reviewer 2 ·

Basic reporting

Clear enough but some improvements in use of literature and context provided would improve it

Experimental design

Minor additional detail in Methods should be added, e.g. seawater temperature and times of coral spawning relative to time of fertilization experiments tarting

Validity of the findings

Straightforward findings. Conclusions could e a little more nuanced.

Additional comments

This manuscript provides new and useful information regarding fertilisation amongst some broadcast spawning Caribbean Scleractinia. The in vitro sperm concentration versus fertilisation success, and the results on gamete co-incubation periods, build on knowledge available for Indo-Pacific species. Similarly, the observations on the interspecific variation in fertilisation success with regards to gamete age are useful. Based on these contributions I recommend publication.
However, a rewrite of the text is required to make better use of available literature, particularly in regards to the overall context provided in the Introduction. Some elaboration of the results in the Discussion would also be beneficial.

Points for Introduction

With regard to revising the Introduction I feel the authors need to capture the general state of knowledge better. For example, the statements beginning on Line 60 that “Currently, coral IVF is practiced following a few rules of thumb” is not really true and at the least requires clarification. It has been 40 years since the prevalence of broadcast spawning among Scleractinians was realised and practitioners have undertaken in vitro culture experiments at multiple locations annually, in both hemispheres, ever since. The original rule of thumb approach was based on evolving practice in the 1980s and 1990s, as it was recognised early on that the sperm concentration versus fertilisation pattern followed a classic S-shaped curve, with fertilisation rates typically falling rapidly at sperm concentration below 104-105/ml. Hence the overwhelming majority of larval culture studies in the last few decades have used a concentration of around 106-107 (e.g. see papers listed in Figure 6; Nakamura et al, 2011, etc, etc). A notable exception has been toxicity-related experiments where ≤106/ml sperm concentrations have been used in order to potentially limit egg-sperm encounter rates and improve the sensitivity of the assay, e.g. Negri & Heyward (2001). This was based on the observed decline in fertilisation rates, as seen on any of the S-shaped fertilisation curves related to sperm concentration and contact time, particularly if sperm concentrations dropped below 105 (see for example the curves in Ricardo et al, 2016 or Buccheri et al, 2023). Hence, nearly all practitioners seeking to maximize fertilisation success have opted for a saturation sperm concentration of ≥106/ml and documented that. Furthermore that concentration has been adopted in major guidelines for production of coral larvae (eg. Guest et al, 2010). Overall the literature has suggested that sperm concentration of 106-107/ml is optimal, yielding maximum fertilisation but moderating risks of polyspermy and reduced water quality (Guest et al, 2010). Hence I feel the authors give the wrong impression with the statement indicating current practice is a “rule of thumb” approach. I recommend the text be revised to clarify and make better use of the literature.
Similarly, there should be some consideration of earlier literature related to gamete age and viability, e.g. Heyward and Babcock, (1986), given oocyte viability in this study was also consistent with early work done on Pacific spawners at 4-6 hours and up to 7 hours.

Points for Discussion

These results, together with other studies, including those listed in Figure 6, suggest that 106/ml is a useful trade-off, delivering maximum fertilisation with lower risk of polyspermy and deleterious water quality that can be associated with higher sperm concentrations. So, for simple larval propagation the “rule of thumb”, for both Indo-Pacific and Caribbean species seems to remain a working sperm dilution in the 106-107 range. I think the authors should point this commonality out explicitly, but then elaborate on scenarios where higher or lower sperm concentrations may be beneficial. For example, long delays in processing gametes from multiple coral species might mean it is several hours before a particular cross-fertilisation trial can be started. In that case, might there be benefits in maintaining live sperm at lower concentrations to improve sperm viability etc? Similarly, are there conditions under which the higher sperm concentrations are preferable, for example as shown by Ricardo et al(2016) in compensating for sediment.
Some elaboration on the Co-incubation period discussion is also warranted. Firstly, some discussion of whether the recorded effective time periods reflect the time required for sperm and eggs to meet and fertilisation to occur, or simply the time for the gametes to meet. For example, it may be that sperm find eggs and remain in proximity or temporarily attached, before locating the receptor site for syngamy to occur. It is normal, when observing gametes in vitro, to see many sperm around and on the oocyte surface. Furthermore, coral oocytes and indeed the egg-sperm bundles, are known to have mucous envelopes (e.g. Padilla-Gamiño et al, 2011; Ricardo et al, 2016; de Cruz et al, 2022; Valente et al, 2024). Combined with the fact that oocytes undergo some changes in shape when they escape from the bundles, it may be that the oocytes themselves require varying amounts of time before they are completely fertilizable. Such factors and presence of mucous layers to varying degrees may also contribute to some species specific differences, such as those you have documented with O. faveolata. Hence I feel you could elaborate a bit more in the Discussion.


Minor specific comments

More detail needs to be given on the timing of gamete collection from the field in relation to the actual timing of coral spawning. This will provide a more accurate age of the gametes PS when the crosses were actually made.
Water temperature in the sea and in vitro should be documented, given thermal plasticity in larval development has been documented and gamete viability might also be affected.
Please add information on oocyte diameters for the four species.
Document any observations if anything different was observed among the species, in terms of bundle size, cohesiveness, mucous, ooctye appearance and so on.

Literature cited in this review
da Cruz, N.O., Galuppo, A.G., Silva, A.G. et al. Assessment of viability in coral oocytes: a biochemical approach to achieve reliable assays. Mar Biol 169, 96 (2022). https://doi.org/10.1007/s00227-022-04086-z
Guest, J. et al (2010). Rearing coral larvae for reef rehabilitation. Pages 73-98 in Edwards, A.J. (ed.) (2010). Reef Rehabilitation Manual. Coral Reef Targeted Research & Capacity Building for Management Program: St Lucia, Australia. ii + 166 pp. (see in particular p86)
Heyward, A.J., Babcock, R.C. Self- and cross-fertilization in scleractinian corals. Mar. Biol. 90, 191–195 (1986). https://doi.org/10.1007/BF00569127
Nakamura R, Ando W, Yamamoto H, Kitano M and others (2011) Corals mass-cultured from eggs and transplanted as juveniles to their native, remote coral reef. Mar Ecol Prog Ser 436:161-168. https://doi.org/10.3354/meps09257
Padilla-Gamiño, J.L., Weatherby, T.M., Waller, R.G. et al. Formation and structural organization of the egg–sperm bundle of the scleractinian coral Montipora capitata . Coral Reefs 30, 371–380 (2011). https://doi.org/10.1007/s00338-010-0700-8
Ricardo, G., Jones, R., Clode, P. et al. Suspended sediments limit coral sperm availability. Sci Rep 5, 18084 (2016). https://doi.org/10.1038/srep18084
Ricardo, G., Jones, R., Negri, A. et al. That sinking feeling: Suspended sediments can prevent the ascent of coral egg bundles. Sci Rep 6, 21567 (2016). https://doi.org/10.1038/srep21567
Wanderson Valente, Cláudia Kelly Fernandes da Cruz, Jener Alexandre Sampaio Zuanon, Gleide Fernandes de Avelar, Leandro Godoy (2024).Ultrastructural evaluation of the oocytes and spermatozoa of the scleractinian coral Mussismilia harttii. Tissue and Cell,Volume 90, https://doi.org/10.1016/j.tice.2024.102469

Reviewer 3 ·

Basic reporting

The paper is well written with clear English.

The literature covered was sufficient. There were a few parts of the Discussion that could have done with more interpretation that I've outlined below.

The Figures were nice except Fig. 3-6 which requires more details including adding raw data points and replacing the connecting line.

The raw data is shared but I encourage the authors to replace % fertilisation with the actual total embryos and total eggs if they want this data to be used in others work e.g. meta-analyses.

Experimental design

The Methods are sufficient, but requires some extra details in the text (see comments below).

Validity of the findings

Generally satisfactory.

I've made a comment about how there is no replication across spawning months, and that this can lead to changes in the trends observed here. To acquire this is often not feasibly possible, but it does need a caveat in the text.

Additional comments

Bennett et al. PeerJ review
Overall comments
This is a great paper that provides useful information about fertilisation kinetics of four species of corals that are frequently used in restoration and management initiatives in the Caribbean. It presents foundational knowledge of IVF thresholds that scientists can use to optimise lab-based coral reproduction procedures. The authors also take this a step further and present an updated synthesis of all other available fertilisation information in the literature which is useful for comparing results across species and locations. This paper will contribute nicely to the coral reproduction literature, which has been largely overlooked in many cases but is crucial to the advancement of reproductive biology and restoration science.
Main feedback
This paper does require revisions to provide more details to the methodology and discussion sections. It is lacking some specific information regarding experimental procedures that would be useful for replication by other scientists in the future. The paper would also benefit from additional details in the discussion section, particularly comparing the experimental results to observed trends in the literature. And explaining why some of the experimental results have been observed, based on gamete properties, specific differences across species, etc. A more in-depth literature discussion would provide useful context to support the findings.
An area to caveat is that the trends observed may be spawning specific i.e. if the experiment was repeated in another spawning month/year, or the adult corals were collected from a different location, the eggs may require more or less sperm to reach maximum fertilisation. This is what we have seen in experiments, and is supported by Nozawa 2015.
The statistics and curve fitting are generally of lower quality. I’m not concerned that most of the results do not hold because of the large number of samples per replicate but I do request that the authors replace connecting lines with straight lines, and note in the Figure captions that these are to visually guide the reader (as opposed to be model fits), I explain this more below. I also request that raw data per replicate are presented on the plot.

Line-specific comments
Line 55 – Great point. Not many ppl realise high fertilisation also means a healthy culture.
Line 62 – I’m surprised ‘ensuring genetic diversity’ in not listed as a primary reason.
Lin e – 91. Note that Willis likely increased the longevity of the sperm through aeration
Line 118 - What dates and times did spawning occur?
Line 122 - How long before spawning were hoods put on? i.e. to avoid stress to colonies
Line 132 - How were the egg and sperm donors chosen?? Aka how did the authors choose which adult colonies to use? How many colonies were being monitored?
Line 138 – I understand the feasibility of not doing more than one egg donor – but good that the authors acknowledged that this could have an effect on results due to egg variability.
How long did it take for all colonies to spawn, and for divers to surface, then to return to the research centre? Rough estimate is okay but would be useful to know a temporal outline of the spawning nights – this is relevant for bundle breakage times, etc.
Line 173 - Some more details in the methods of the first 2 assays would be useful – i.e. how much sperm from each colony, how much total (after pooled), how much was used for each dilution (for assay 1), and for each trial with eggs, etc.
Roughly how many eggs were in each trial? And how many were counted to assess fertilisation success in each treatment? Add how many reps for each assay. You say 3-6 earlier, but it looks like mostly 3 in your raw data.
Line 177 - Maybe better description of the dilutions instead of “1/10” or at least describe what this means in more detail.
Line 180 – Self fertilisation is where the sperm for an individual fertilises an egg from the same individual, so to test this you would need to not wash off the sperm. If you didn’t do this, you should say that the effect of self-fertilisation was not assessed.
Technically, there are types of parthenogenesis (gynogenesis) where the sperm can also stimulate the egg to form an embryo but without fusion.
Line 190 – Were the gametes left static or were they aerated or agitated during this period?
Line 210 - How many hours post spawning was this 64-cell stage photographing and counting done? How many eggs were counted per treatment? Were there any fragments in the sample. 64-cell stage is generally delicate.
Line 215 - More detail in the data analysis section would be beneficial.
Line 284-285– Could use more discussion into why sperm concentration requirements might be different across species. Can discuss previous literature relating egg size to fertilisation, etc.
Line 287 – Could benefit from clearer description of why polyspermy/other negative effects at high sperm concs may not be observed here (i.e. polyspermy blocks, other findings from the literature). There is some info at line 337-338 – consider moving up to the sperm conc section.
Line 287 - This section could note that the super high sperm concentrations are not biologically relevant in most natural spawning scenarios? So this is more relevant for lab based scenarios
Line 299 - Gamete age paragraph does well with bringing in some previous literature to explain the trends you are observing.
Line 307 – Also note that another mechanism (although unproven if actually a limiting factor for corals) is the release of the first polar body. The oocyte is considered infertile before its release. Heyward and Babcock 1986.
Line 308-309 - rewrite this line – doesn’t seem like the “if” belongs there.
Line 311 – this paragraph uses previous literature nicely!
Line 335 – Could add a sentence to note that sperm motility and gamete recognition capabilities vary across species – so we can expect the sensitivity of fertilisation to different incubation times will be variable across species.
Line 340-346. Needs more references. E.g Levitan, Yund etc.
Line 341 – change to gamete encounters
Line 343 – thus*
Line 343 – Should add explanation why – i.e. due to hydrodynamic mixing that gametes are exposed to on spawning nights. There is some literature for other spawning inverts that discusses this.
Line 356 – great that the authors are discussing individual incompatibility because it is often overlooked in the coral literature! But another mitigation strategy for minimising the likelihoods of incompatibilities negatively impacting IVF would also be to add more adult sperm and egg donors, though not always feasible depending on the availability of gametes/gravid adults. Authors should add a sentence or two to note this, especially because it directly relates to their experimental design.
Line 369 – The authors could also add a sentence in the discussion or conclusions section to tie these results to in situ spawning processes. The paper is specifically focused on IVF and lab-based work, but the results will still be useful to managers of in situ coral populations. A sentence or two about how the three parameters you manipulated are affected by declines in natural spawning populations, etc. that we are seeing in the Anthropocene will give broader context to the work.

Figures:
Figure 1 is excellent. It provides a clear illustration of the experimental methodology. The “4 species” note in the step 1 panel may be unnecessary. And the text font size would benefit from enlargement throughout.
Would also recommend changing the species colours to a colour-blind friendly palette. Some people struggle to distinguish between red and green.
Figure 2 – Great
Figure 3-6 and stats. Typically, it is more defendable and useful to treat grouped continuous data (i.e. sperm concentration) as continuous, to adjust for overdispersion (this type of data is often variable), and to weight each sample based on size (e.g. some replicates might contain twice as many eggs initially than others, so they should have a greater weighting). Something like a GLMM or a GAMM can handle this. I’m not too concerned about this because of the large sample size and effect size.

The use of the polynomial? connecting line is a bit misleading as it indicates a model fit. If you want to keep it, you should describe the model, add confidence intervals, and interpret the data based on the model fit. Otherwise, replace with a straight connecting line, and note that this is just to guide the readers visually in each figure caption.
Figure Captions: Add more details to the captions e.g. species, replicates per treatment etc. A Figure caption should be stand-alone.

Reviewer 4 ·

Basic reporting

This study examines fertilization kinetics, including sperm concentration and gamete aging, in several previously unexamined coral species. The protocols applied by the authors are classic and well-established. The authors used multiple sires to fertilize eggs, thereby ensuring a more robust connection to producing genetically diverse larvae. This is essential for a practical understanding of coral sexual reproduction.

Introduction
Line 98-100: This statement is crucial. Other studies refer to IVF if the authors consider fertilization in the tubes as "IVF." The authors examined IVF with multiple sires crossing with eggs from a single dam. It would be helpful to emphasize these points to highlight the importance of your approach.

Materials and Methods
Lines 138-144: These sentences may not be necessary. I suggest the authors either eliminate them or clarify that they could not conduct all combinations to account for the variability in crossing rates among the dams and sires.

Line 157-158: "We subsequently pooled sperm from all three donors in equal parts to ensure each sire was approximately equally represented in the sperm pool (Fig. 2)."
Consider revising to: "We subsequently mixed equal volumes of sperm from all three donors to ensure that each sire was approximately equally represented in the sperm pool."

Discussion
Line 311-316: "Together, these findings present restoration opportunities whereby gametes could be collected and later combined from sites representing more distant and distinct habitat types, thus enhancing potentially adaptive genetic diversity in larval cultures (Baums et al., 2019)."

While I agree with this statement, it appears to contradict the study's findings that higher sperm concentration is optimal for fertilization in the examined species. In contrast, Kitanobo et al. (2022) showed that fertilization in situ may occur at much lower sperm concentrations (10^4 sperm/ml). I recommend discussing the limitations of applying in vitro findings to in situ conditions, as this could affect the interpretation of restoration strategies.

Experimental design

No comment

Validity of the findings

This study provides essential information about coral reproduction.

Reviewer 5 ·

Basic reporting

.

Experimental design

.

Validity of the findings

.

Additional comments

I have reviewed the manuscript by Bennett et al, describing Optimizing in vitro fertilization in four Caribbean coral species. Overall, this paper makes a significant contribution to the field of coral restoration by optimizing IVF protocols for important Caribbean coral species. While the study is well designed and relevant, addressing issues of biological replication, sperm quantification, and statistical analysis would improve the robustness and applicability of the findings. Further research on assessing long-term gamete viability and embryo health post-fertilization is strongly recommended. I recommend that this paper be published with minor revisions to strengthen the manuscript as follows:
Line 132… we selected three parents as sperm-donors and a separate parent as egg-donor…. The coral parents was restricted to only one colony for male and one colony for female, which lacks biological replication and limits the generalizability of the findings. we recommend using multiple parents from colonies with different population diversity, e.g. from locations other than the study site.
Line 180, parthenogenesis, self-fertilization, and/or sample contamination with non self sperm…. If possible, this control section is interesting to see the results and expansion of the discussion including specific results and their implications to strengthen the validity and clarity of this phenomenon on an in vitro scale
Line 217, used one-way Kruskal-Wallis non-parametric tests...., it is recommended to increase statistical analysis using the General Liner Model (GLM) also to provide results and insight into the interaction between the variables used and their effects on fertilization
Line 170-171, under a phase contrast optical microscope ….it would be valuable to present the fertilization results of each type of coral test from this microscope photo in a figure.
Line 277-280, Sperm concentration was a strong driver of fertilization success…. this sentence should be moved to the results sub-chapter.

---

## Round 0.2 · Minor Revisions

Dear Dr. Chamberland,

Your paper has been reviewed by four experts in the field, all of whom agree that your research provides valuable insights for improving Caribbean reef restoration and is deserving of publication. However, one of the reviewers has offered some suggestions that we kindly request you address before proceeding with publication.

Please revise the manuscript accordingly, incorporating the reviewer’s feedback, and resubmit it for further consideration.

Thank you for your important contribution to the field, and we look forward to receiving your revised manuscript.

Best regards,

Guilherme

Reviewer 2 ·

Basic reporting

Paper reads well, provides valuable information and should be published.

Experimental design

Adequate

Validity of the findings

Useful findings for Caribbean corals and of relevance to coral reproductive studies elsewhere

Additional comments

Some minor rewriting is suggested to improve clarity, mainly in the Abstract.
An additional very relevant and more recent reference provided for the authors to consider.

Specific comments as follows;

L39 suggest changing “stresses” to “elevates”
L40-41 suggest changing ”and climate change” to “and broader climate change effects”
L48 for clarity recommend you introduce the term “controlled or hatchery-based” into this sentence, something like” controlled sexual propagation” given numerous practitioners undertake sexual coral propagation by harvesting wild coral embryos and larvae, often aggregated into surface slicks, but usually hours after spawning so they do not collect gametes or undertake in vitro fertilisation.
L53 onwards Try and mention the idea of increasing larval density in culture facilities being important. This study seeks to optimise fertilisation and maintain water quality so that the efficiency of larval propagation is improved. The ideas are good, but the rational is important because you are talking about “intensive “ aquaculture in fixed volume facilities that are expensive to build and run. So you are trying to get more bang for the buck by maximizing production form a fixed volume/cost system.
L87-90 A more recent paper on coral fertilization by Severati et al, 2024 provides detailed information on IVF, sperm concentration, timing etc, and is very relevant to this manuscript in a general sense. It contains data specifically relevant to this section of the manuscript.
Severati et al (2024).The AutoSpawner system - Automated ex situ spawning and fertilisation of corals for reef restoration. Journal of Environmental Management, Volume 366. ttps://doi.org/10.1016/j.jenvman.2024.121886.

L120 Suggest inserting “Detailed” before “information”, given some information on IVF for those species is known
L447-453 I understand what you are trying to say, but suggest it could be worded better. This section needs to clarify that the issue is colony density, not population size. Synchronous spawning of corals means that gamete encounter and cross-fertilisation is more likely between nearest neighbours in the initial time period after gamete release. So the distance between neighbouring conspecifics is a key issue if colony density decreases. Given gamete viability of at least 4 hours it is true that gametes can disperse widely, but with both egg and sperm dilution come into play, e.g, Oliver & Babcock, 1992.
L465 “damage to developing embryos” may need to be briefly explained here. The concern is excessive physical force when washing the embryos, hence gentle washing e.g., this MS, L89, had become accepted practice. The fragility of early stage dividing embryos to robust water movement was highlighted by Heyward & Negri (Science. 2012 Mar 2;335(6072):1064. doi: 10.1126/science.1216055.), but practices have evolved over recent years to manage washing processes before first cleavage. The Autospawner system (Serverti et al, 2024) demonstrates that oocytes and pre-cleavage zygotes can tolerate quite intense washing.

Reviewer 3 ·

Basic reporting

no comment

Experimental design

no comment

Validity of the findings

no comment

Additional comments

This revision has satisfactory addressed my comments and I look forward to seeing it out.

Reviewer 4 ·

Basic reporting

Thank you for revising according to my comments.

Experimental design

no comment

Validity of the findings

no comment

Additional comments

no comment

Reviewer 5 ·

Basic reporting

Article improvements are appropriate and have followed an improvement review and editorial criteria

Experimental design

Article improvements are appropriate and have followed an improvement review

Validity of the findings

no comment

Additional comments

This article is worthy of being accepted and published, it is in accordance with the editorial writing published by Peer J. Substantially, this article provides new knowledge about coral reproductive biology with clear novelty and contributes to efforts to preserve, conserve and utilize corals properly and sustainably.

---

## Round 0.3 · accepted · Accept

Thank you for carefully addressing the reviewer's suggestions. Congratulations on your great work!